# Use of Selected Carbon Nanoparticles as Melittin Carriers for MCF-7 and MDA-MB-231 Human Breast Cancer Cells

**DOI:** 10.3390/ma13010090

**Published:** 2019-12-23

**Authors:** Karolina Daniluk, Marta Kutwin, Marta Grodzik, Mateusz Wierzbicki, Barbara Strojny, Jarosław Szczepaniak, Jaśmina Bałaban, Malwina Sosnowska, Andre Chwalibog, Ewa Sawosz, Sławomir Jaworski

**Affiliations:** 1Department of Nanobiotechnology and Experimental Ecology, Institute of Biology, Warsaw University of Life Sciences, 02-787 Warsaw, Poland; karolina_daniluk@sggw.pl (K.D.); marta_prasek@sggw.pl (M.K.); marta_grodzik@sggw.pl (M.G.); mateusz_wierzbicki@sggw.pl (M.W.); barbara_strojny@sggw.pl (B.S.); jaroslaw_szczepaniak@sggw.pl (J.S.); jasmina_balaban@sggw.pl (J.B.); malwina_sosnowska@sggw.pl (M.S.); ewa_sawosz@sggw.pl (E.S.); 2Department of Veterinary and Animal Sciences, University of Copenhagen, Groennegaardsvej 3, 1870 Frederiksberg, Denmark; ach@sund.ku.dk

**Keywords:** breast cancer, melittin, carbon nanoparticles, drug delivery, nanographene oxide, graphene, nanodiamond, cell death

## Abstract

Despite advanced techniques in medicine, breast cancer caused the deaths of 627,000 women in 2018. Melittin, the main component of bee venom, has lytic properties for many types of cells, including cancer cells. To increase its toxic effect, carbon nanoparticles, graphene oxide, pristine graphene, and diamond were used as carriers of melittin to breast cancer cells. To date, the effects of carbon nanoparticles as carriers of melittin on cancer cells have not been studied. The present study was carried out on MCF-7 and MDA-MB-231 cell lines. The investigation consisted of structural analysis of complexes using transmission electron microscopy, zeta potential measurements, evaluation of cell morphology, assessment of cell viability and membrane integrity, investigation of reactive oxygen species production, and investigation of mitochondrial membrane potential. Cell death was examined by flow cytometry and a membrane test for 43 apoptotic proteins. The results indicate that melittin complex with nanographene oxide has a stronger toxic effect on breast cancer cells than melittin alone. Moreover, nanodiamonds can protect cells against the lytic effects of melittin. All complexes reduced, but not completely eliminated the level of necrosis, compared to melittin. Thus, results suggest that the use of carbon nanoparticles as carriers for melittin may find use in medicine in the future.

## 1. Introduction

According to the World Health Organization, breast cancer is the most frequent cancer in women, impacting 2.1 million women each year and also causes the greatest number of cancer-related deaths among women. In 2018, it was estimated that 627,000 women died from breast cancer, which accounts for approximately 15% of all cancer deaths among women. While breast cancer rates are higher among women in more developed regions, the rates are increasing in nearly all regions globally [1]. Developmental advances in medicine have provided us with many types of treatment for this type of cancer, including surgery to remove the whole breast or the tumor and tissues around it, and radiation therapy, which uses high-energy waves to destroy cancer cells. Other treatments destroy or control cancer cells all over the body; with chemotherapy, drugs are used to kill cancer cells. As these powerful drugs fight the disease, they can also cause side effects, such as nausea, hair loss, early menopause, hot flashes, and fatigue. During hormone therapy, drugs are used to prevent hormones, especially estrogen, from fueling the growth of breast cancer cells. Targeted therapy prompts the body’s immune system to destroy cancer. Despite this, many women and men die of this type of cancer. Therefore, new methods of treatment are constantly needed, often with the use of natural sources, and one among these methods is melittin (MEL).

MEL, the main biologically active component in bee venom, comprises approximately 40–50% of the venom. It is a linear peptide consisting of 26 amino acid with water-soluble and amphipathic properties [2]. Many studies have shown the biological effects of MEL as antibacterial, antiviral, antifungal, and anti-parasitic; and suggest that MEL has non-selective cytolytic activity, acting physically and chemically to disrupt all prokaryotic and eukaryotic cell membranes (Figure 1) [3,4,5]. The antibacterial activity against *Xanthomonas oryzae* has been demonstrated as a consequence of the interaction of MEL with the bacterial cell wall, leading to disruption of the cytoplasmic membrane by formatting channels and leakage of cytoplasm. The same study suggests that MEL may have the ability to bind the RNA or DNA in the cytoplasm leading to bacterial cell death. The antiviral activity of this peptide has been proven in an influenza virus model. MEL displays virucidal activity, disrupting viral envelopes [6].

Among the known zootoxins, MEL also has anticancer properties, which has been demonstrated for many cancers. Previous researches showed that MEL inhibits cell proliferation through an increase in death receptor expressions and by the induction of apoptotic cell death in a dose dependent manner in the human ovarian cancer cells [7]. Additionally, a previous study suggested that natural peptide from bee venom can induce apoptotic cell death by inhibiting the STAT3 pathway and causing an increase in death receptor (DR) 3, DR4, and DR6 expression. Another study showed that MEL has a potential role as a therapeutic agent in prostate treatment. The peptide alleviated inflammation by suppressing cyclooxygenase 2 expression in rats. Tests carried out on a glioblastoma multiforme cell line showed a devastating effect of MEL on the glioma cells. MEL causes disintegration of cell membranes and induces cell death by apoptosis and less by necrosis [7,8,9].

Although the potential utility of MEL as cancer chemotherapeutic has long been studied, its rapid degradation in the blood and non-specific lytic activity of cells is a big challenge [10]. This peptide, after intravenous injection, causes severe toxic reactions, such as hemolysis, which is a limitation of its widespread use in the treatment of cancer [11]. Despite the knowledge of the mechanism of MEL interaction with biological membranes, the molecular effects of its action vary depending on the type of cells and the activity depends on the tumor microenvironment. Due to the non-specific activity of MEL, it is necessary to use a delivery system for this peptide. Nanomaterials appear to be promising as MEL carriers because of their properties and sizes. 

Nanotechnology is a fast-developing multifaceted field, which is used in increasingly newer areas of science. Recently, nanomaterials and nanoparticles have found many applications in the field of nanomedicine, particularly in drug delivery systems (Figure 2). Over the past decade, many studies have focused on the potential use of graphene as a carrier for the targeted delivery of drugs for the diagnosis and therapy of cancer. Recent results showed that the use of graphene as a drug carrier resulted in increased solubility in water and allows for a higher drug concentration without any accumulation of nanoparticles [12]. Nanographene oxide (nGO) has several features, that indicate the possibility of using it as a carrier to cancer cells, like biocompatibility, high ability to adsorb drugs, as well as easy surface functionalization [13]. However, its biological properties vary depending on the concentration and size of the nanoparticles [14,15]. Its properties have shown potential use as a carrier of doxorubicin and siRNA to drug-resistant cancer cells [16]. Furthermore, a nanodiamond (ND) was used as a slow drug release system in contact lenses of people with glaucoma [17].

The nanoparticles are promising carriers of therapeutic agents because of the mechanism of their uptake by cells called “internalization” [18], which is another term for endocytosis, in which molecules such as proteins are engulfed by the cell membrane and transported into the cell from their external environment. The examined nanoparticles did not show strong toxicity, but effectively showed deregulated cell migration. ND was effectively taken up by cells, whereas nGO and graphene (GN) strongly interacted with the cell surface [19].

In recent years, apoptosis is a known desirable effect of newly sought therapies that generally does not cause inflammation and is known as a less noxious type of cell death. Recently, the influence of MEL on the regulation of apoptosis in many types of cancers have been extensively studied. It was observed that MEL activates caspases in leukemia, melanoma, prostate, and cervical cells [20,21,22]. Further, it was proven that small peptides from bee venom downregulate the signal pathway causing apoptosis in leukemic cells. Results from the same study indicate that apoptosis caused by MEL connect with activation of caspase-3, downregulation of Bcl-2, and reduced expression of the inhibitor of apoptosis protein. Another study reported that MEL increased protein expression of cytochrome C, Endo G, and AIF, and downregulated the Smac/Diablo protein. However, the effects differ for the different cell types. MEL exhibits necrotic cytotoxicity in gastrointestinal cells [23]. Therefore, it is important for each type of cancer to examine the effect of this peptide. The challenge in its use is the lytic activity of MEL in relation to all biological membranes that can be manipulated using different particles. Carbon nanoparticles, as materials taken up by cells, in a complex with MEL showing the lytic activity can increase the efficiency of destruction of extracellular and intracellular membranes and activate programmed cell death in breast adenocarcinoma cells. 

We achieved promising results in our previous studies with glioma cell lines treated with MEL and MEL conjugated with carbon nanoparticles [24]. Results of the work prove the effectiveness of carbon nanoparticles as nanocarriers, especially ND, GN and nGO. They effectively transport the targeted MEL and help with adhesion and internalization of it to the glioma cells. We hypothesized that carbon nanoparticles, as factors that adhere easily to cell surfaces and are absorbed by the cell, can increase the local toxic effect of MEL by increasing uptake of MEL with carbon nanoparticles by breast cancer cells. The objective of this study was to determine the use of carbon nanoparticles as MEL carriers for breast cancer cells on in vitro stage by studying the toxicity, dose dependence, and the proapoptotic and necrotic activities to examine feasibility of use for clinical breast cancer treatment.

## 2. Materials and Methods

### 2.1. Melittin and Carbon Nanoparticles

Pure MEL peptide was obtained from Sigma-Aldrich (purity ≥85%; Munich, Germany) in a powder form and was dissolved in ultrapure water. GN (thickness: 6–8 nm; purity >99.5%) and ND (thickness 3–4 nm; purity >95%) powders were purchased from the SkySpring Nanomaterials (Houston, TX, USA) and nGO (thickness <2 nm; purity 98%; diameter: 8–15 um) powders were purchased from Institute of Electronic Materials Technology in Warsaw, Poland. To select the concentrations for the study on complexes, the viability (XTT test) and mortality (PrestoBlue test) of breast adenocarcinoma cells (MCF-7 and MDA-MB-231) were measured for five different concentrations of hydrocolloids of each nanoparticles (5, 10, 20, 50, 100 mg/L), MEL (1, 2, 5, 10, 20 mg/L) and the control. Cells cultured without the addition of nanoparticles or MEL, but with the same volume of media, were used as the control group.

### 2.2. Cell Lines

Human breast adenocarcinoma MCF-7 and MDA-MB-231 cell lines were obtained from American Type Culture Collection (Manassas, VA, USA) and maintained in Dulbecco’s modified Eagle’s culture medium containing 10% fetal bovine serum (Life Technologies, Houston, TX, USA), 1% penicillin, and streptomycin (Life Technologies) at 37 °C in a humidified atmosphere of 5% CO_2_/95% air in a NuAire DH AutoFlow CO_2_ Air-Jacketed Incubator (Plymouth, MN, USA).

### 2.3. Preparation and Characterization of Complexes

MEL was added to each type of nanoparticle to obtain three different complexes in concentrations of 20 μg/mL for nanoparticles and 10 μg/mL for MEL. Then, the complexes were incubated at room temperature and vortexed for 15 min. MEL, three different nanoparticles, and complexes of nanoparticles were analyzed using a transmission electron microscope (TEM) JEM-1220 (JEOL, Tokyo, Japan) at 80 KeV, with a Morada 11-megapixel camera (Olympus Soft Imaging Solutions, Münster, Germany). Samples for the TEM were prepared by placing droplets of hydrocolloids on to Formvar-coated copper grids (Agar Scientific, Stansted, UK). Immediately after drying the droplets in the room temperature, the grids were inserted into the TEM.

Zeta potential of nanoparticles, MEL, and complexes colloids was measured by the laser dynamic scattering-electrophoretic method with Smoluchowski approximation using a Zetasizer Nano ZS, model ZEN3500 (Malvern Instruments, Malvern, UK). Each sample was measured after 120 s of stabilization at 25 °C. Nanoparticles were sonicated for 30 min and the zeta potential was immediately measured. The zeta potential of the complexes was measured immediately after 15 min incubation of MEL with nanoparticles.

### 2.4. Morphology Analysis

After 24 h of exposure to the MEL and MEL–nanoparticles complexes, the medium was removed, the cells were stained using the May–Grünwald–Giemsa (Sigma-Aldrich) method, and their morphology was investigated using a CKX 41 light microscope (Olympus, Tokyo, Japan). Images were captured using a ProgRes c12 camera (Jenoptik, Jena, Germany). Cells cultured without the addition of MEL–nanoparticles or MEL, but with the same volume of media, were used as the control group.

### 2.5. Cell Mortality Assay

Cell mortality was evaluated using a PrestoBlue™ Cell Viability Assay (Life Technologies, Taastrup, Denmark). The PrestoBlue™ reagent is quickly reduced by metabolically active cells, providing a quantitative measure of viability and cytotoxicity. MCF-7 and MDA-MB-231 cells were plated in 96-well plates (5 × 10^3^ cells per well) and incubated for 24 h. Then, the medium was removed, and hydrocolloids of each nanoparticle (5, 10, 20, 50, 100 mg/L) and MEL (1, 2, 5, 10, 20 mg/L) were introduced to the cells. After 24 h, 10 µL of PrestoBlue™ reagent was added to each well and incubated for an additional 2 h at 37 °C. After incubation, the plate was centrifuged (5 min, 1000 RPM) and 100 mL was transferred to a new plate. The optical density of each well was recorded at 570 nm on an enzyme-linked immunosorbent assay reader (Infinite M200, Tecan, Durham, NC, USA). Cell viability was expressed as the percentage (ODtest− Dblank)/(ODcontrol−ODblank), where “ODtest” is the optical density of cells exposed to nanoparticles and MEL, “ODcontrol” is the optical density of the control sample, and “ODblank” is the optical density of wells and without cells. The test was replicated five times for each group.

### 2.6. Cell Metabolic Activity

The metabolic rates of the MCF-7 and MDA-MB-231 cells were evaluated using a 2.3-Bis-(2-methoxy-4-nitro-5-sulfophenyl)-2H-tetrazolium-5-carboxyanilide salt (XTT)-based cell proliferation assay kit (Merck, Darmstradt, Germany). The assay is based on the extracellular reduction of XTT by NADH produced in the mitochondria. XTT is reduced to formazan only by metabolically active cells. The formation of formazan causes a change in the color of the fluid and enables reading by measuring absorbance.

MCF-7 and MDA-MB-231 cells were plated in 96-well plates (5 × 10^4^ cells per well) and incubated for 24 h. Then, the medium was removed, and hydrocolloids of each nanoparticle (5, 10, 20, 50, 100 mg/L) and MEL (1, 2, 5, 10, 20 mg/L) were introduced to the cells. This test was also carried out at a later stage examining the effect of complexes on cells. After 24 h incubation, 50 μL of XTT solution was added to each well and incubated for an additional 4 h at 37 °C. After incubation, the plate was centrifuged (5 min, 1000 RPM) and 100 mL was transferred to a new plate. The optical density (OD) of each well was recorded at 450 nm on an enzyme-linked immunosorbent assay reader (Infinite M200, Tecan, Durham, NC, USA). Cell viability was expressed as the percentage of (ODtest−ODblank)/(ODcontrol−ODblank), where “ODtest” is the optical density of cells exposed to MEL and nanoparticles, “ODcontrol” is the optical density of the control sample, and “ODblank” is the optical density of wells without cells. The same procedure was performed for the studied complexes in concentrations of 20 μg/mL for nanoparticles and 10 μg/mL for MEL. The test was replicated five times for each group.

### 2.7. Membrane Integrity

A lactic dehydrogenase (LDH) test (performed using an LDH-based in vitro toxicology assay kit, Sigma-Aldrich) was used to evaluate cell membrane integrity. The test is based on reduction of NAD to NADH by LDH, which then interacts with a specific probe to produce a color. The change in the color of the medium, relative to the control group, indicates the leakage of LDH from the cells and damage to the cell membranes.

MCF-7 and MDA-MB-231 cells were plated in 96-well plates (5 × 10^3^ cells per well) and incubated for 24 h. Afterward, the medium was removed, and hydrocolloids of complexes and MEL were introduced to the cells. After incubation, half the volume of the culture medium was removed. A total of 100 μL of the LDH assay mixture was added to each well. The plate was covered and incubated for 20 min at room temperature. The OD was recorded as outlined, and the LDH leakage was expressed as the percentage of OD. The resulting reduced nicotinamide adenine dinucleotide (NADH+) was utilized in the stoichiometric conversion of a tetrazolium dye. When cell-free aliquots of the medium from cultures, with different treatments, were assayed, the amount of LDH activity could be used as an indicator of relative cell viability, as well as a function of membrane integrity. If the membrane is damaged, intracellular LDH molecules are released into the culture medium. Thus, the LDH level in the medium indicates cell membrane integrity. The test was replicated five times for each group.

### 2.8. Intracellular Reactive Oxygen Species (ROS) Level

Intracellular ROS level was assessed using a fluorometric intracellular ROS kit (Sigma, St Louis, MO, USA). MCF-7 and MDA-MB-231 cells were plated in 96-well plates (1 × 10^4^ cells per well) and incubated in a 5% CO_2_, 37 °C incubator for 1 h. Afterwards, the Master Reaction Mix was added into the cell plate, immediately after which fluorescence intensity was measured at λex520 nm/λem605 nm with a microplate reader and in 5-min intervals. The test was replicated five times for each group.

### 2.9. Potential of Mitochondrial Membrane

JC-10 staining was used to evaluate mitochondrial membrane integrity. The test is based on a dye that selectively enters the mitochondria and reversibly changes its color from green to orange as membrane potentials increase. It is possible to monitor the polarization changes of mitochondrial membranes caused by membrane damage.

MCF-7 and MDA-MB-231 cells were plated in 96-well plates (5 × 10^3^ cells per well) and incubated for 24 h. Afterwards, the medium was removed, and hydrocolloids of nanoparticles, complexes, and MEL were introduced to the cells. After 24 h of incubation, the staining solution was added to the cells. After 1 h of incubation without light, the stop solution was added. The fluorescence intensity was measured as Ex/Em = 490/525 and 540/590 nm. The test was replicated five times for each group.

### 2.10. Level of Human Apoptosis Proteins

TissueLyser LT (Qiagen, Hilden, Germany) was used to prepare protein extracts. Protein concentrations of tissues lysates were determined using the BCA Protein Assay (Thermo Scientific). The levels of multiple apoptosis protein were examined by the Human Apoptosis Antibody Array - Membrane (ab134001, Life Technologies), prepared for the simultaneous detection of 43 proteins. The following targets could be detected by this array.

Three samples from each group were diluted to a final concentration of 5 μg/μL. The cytokine array was performed according to the manufacturer’s instructions. Chemiluminescence detection was performed using multiple exposure times (30 s to 5 min) with the ChemiDoc1 Imaging System using Quantity One Basic Software (Bio-Rad, Hercules, CA, USA).

### 2.11. Apoptosis/Necrosis Assay

Apoptosis and necrosis were evaluated using the fluorescein isothiocyanate (FITC) Annexin V/Dead Cell Apoptosis Kit (Thermo Scientific, Waltham, MA, USA). The annexin V/PI assay is an effective method for detecting the type of cell death by detecting the location of phosphatidylserine on cell membranes. The detection process is possible by monitoring the phosphatidylserine location by annexin V, which has a strong and specific affinity to phosphatidylserine.

MDA-MB-231 cells were seeded at 2 × 10^5^ cells per well in six-well plates and cultured until 70–80% confluence. After 24 h, the medium was removed and hydrocolloids of complexes (MEL-GN, MEL-ND) and MEL were introduced to the cells. The control sample consisted of cells cultured without the addition of MEL and nanoparticles. After 24-h incubation with MEL and complexes, the cells were trypsinized, harvested, and washed with cold phosphate-buffered saline containing calcium and magnesium ions. After rinsing and centrifugation, each cell pellet was suspended in 100 μL of annexin-binding buffer. Then, 5 μL of FITC-labeled annexin V and 1 μL of propidium iodide (PI) were added to each cell suspension. The cells were incubated at 25 °C for 15 min. Then, 400 μL annexin-binding buffer was added to the cell suspensions, gently mixed, and stored on ice until introduction into the flow cytometer. A total of 10,000 events were recorded per sample. Plots were generated using Flowing Software 2.5.1 (Perttu Terho, Turku, Finland). Fluorescence emission intensity was measured using two FL1 channels for FITC at Em = 530 nm and FL2 for PI at Em > 570 nm.

### 2.12. Statistical Analysis

Data were analyzed using multifactorial and monofactorial analysis of variance with Statgraphics^®^ Plus 4.1 (StatPoint Technologies, Warrenton, VA, USA). The differences between groups were tested using Tukey’s multiple range tests. All mean values are presented with the standard deviation. Differences with *p* < 0.05 were considered significant.

## 3. Results

### 3.1. Concentration of MEL and Nanoparticles

The results from PrestoBlue, XTT tests, and the literature allowed for the selection of appropriate concentrations of MEL and nanoparticles to form complexes. According to the preliminary studies, we chose a concentration of 10 mg/L for MEL and 20 mg/L for GN, nGO, and ND.

### 3.2. Visualization and Stability of Nanoparticles and Complexes

Images from TEM showed that MEL was located between the nanoparticles nGO (Figure 2C) and ND (Figure 2D). In complex MEL-GN (Figure 2B), the peptide was evenly dispersed on GN surface.

The results of the zeta potential measurement were different for the studied nanoparticles and their complexes, with MEL showing that MEL-GN complex had greater stability than MEL alone (Figure 3).

### 3.3. Morphology

Changes in cell morphology after treatments with MEL and complexes were observed for both lines (Figure 4). The cell bodies shriveled, and the cell protuberances were clearly shorter compared to the control group. For MCF-7 and MDA-MB-231 cell line, the use of complexes with GN and GO caused a greater destruction of cell morphology compared to the action of MEL itself. While in the group treated with MEL-ND complex, smaller changes in morphology were observed compared to the action of MEL alone.

### 3.4. Metabolic Activity

XTT test is a colorimetric test used to determine the metabolic activity of cells. The results are presented in Figure 5. The study showed that the MEL-nGO complex significantly reduced the metabolic activity of the MDA-MB-231 and MCF-7 cell lines below 40%, relative to the control.

### 3.5. Membrane Integrity

The leak in LDH to the medium indicates the damage to the cell membranes. The test showed that nGO and MEL-nGO caused a leak of 80% of the total LDH to the medium for MDA-MD-231 cells. ND complex with MEL significantly reduced the LDH level for both tested lines (Figure 6). 

### 3.6. Intracellular ROS Level

The level of intracellular ROS was examined using a sensitive fluorometric microplate test. All complexes were used to increase the levels of intracellular ROS. The most significant differences were observed between MEL-GN and MEL-nGO (Figure 7).

### 3.7. Potential of Mitochondrial Membrane

Mitochondrial membrane potential was tested using JC-10 Assay Kit. Results are presented as the ratio of green to orange fluorescence (Figure 8). Orange fluorescence corresponded to the dye accumulated in the mitochondria, that became possible due to the correct polarization of the membranes. The green fluorescence corresponds to the dye, which is in the cytosol. For both lines, a significant decrease in mitochondrial potential was observed for the group treated with MEL-nGO complex.

### 3.8. Level of Human Apoptosis Proteins

A significant increase in the protein expression was observed for Bax, HTRA, Casp3, and Casp8 in MDA-MB-231 cells treated with MEL-GN (Figure 9). A significant decrease in the expression level was observed for p21 and XIAP in the group treated with MEL-GN.

### 3.9. Apoptosis/Necrosis

Apoptosis assay provides a simple and effective method for detecting one of the earliest events in apoptosis: the externalization of phosphatidylserine (PS) in living cells (Figure 10). Soon after apoptosis is induced, PS is translocated from the inner leaflet of the plasma membrane to the outer leaflet. This assay used Annexin V, which has a strong and specific affinity for PS, for monitoring the PS translocation due to apoptosis. MEL induced cell death mostly occurred by necrosis in MCF-7 (26%) and MDA-MB-231 (37%) cells. The use of selected nanoparticles resulted in an increase in the percentage of living cells relative to that in the MEL-treated group. MEL-nGO complex induced cell death by apoptosis in two cell lines (about 30% for MCF-7; over 23% for MDA-MB-231) and reduced the percentage of necrotic cells by more than half compared to the MEL-treated group. An increase in the percentage of viable cells for both lines was observed for MEL-ND treated groups, which may indicate that ND was surrounding MEL and blocking its action.

## 4. Discussion

From the biological point of view, the main action of MEL is the formation of pores in all biological membranes. Due to its activity, many scientists have tried to use this peptide against cancer. Studies show that MEL causes the degradation of tumor cell membranes with varying intensity depending on the type of cancer. However, its action also includes healthy cells, which is why an effective method of its delivery specifically for cancer cells is constantly sought [25,26,27]. 

Several solutions for MEL delivery have been proposed, including nanotechnology. This study aimed to determine the action of MEL-nanoparticle complexes on MCF-7 and MDA-MB-231 breast cancer cells morphology, metabolic activity, cell membrane integrity, potential of mitochondrial membrane, level of intracellular ROS production, analysis of cell death, and apoptosis proteins level to examine feasibility of use for clinical breast cancer treatment.

Previous studies suggest that the use of carbon nanoparticles as MEL carriers can increase the toxic effect of MEL on breast cancer cells, compared to action by MEL alone [24]. TEM imaging of complexes allowed determination of the location of the MEL in relation to nanoparticles. Results show that nGO and ND didn’t completely surround MEL and the peptide stayed on the outside of the complex with space to interact with the cancer cells. MEL in complex with GN was entirely on the GN flake, which may suggest less interaction of MEL with the cell membrane than nGO-MEL and ND-MEL. The analysis of cell morphology demonstrated that GN- and nGO-MEL complexes have a stronger negative effect on cell structure than MEL itself for both treated cell lines. For both cell lines, the use of complexes with GN and GO caused a greater destruction of cell morphology compared to the action of MEL itself. The results suggest that breast cancer cells could absorb the carbon nanoparticles, reducing the consumption of MEL, relative to a single cell. Microscopic visualization of interactions between carbon nanoparticles and breast cancer cells showed that all tested nanoparticles adhered to the cells. Additionally, this is consistent with the results that were focused on the antimicrobial properties of MEL. MEL hybrids with nGO and GN showed a 20-fold increase in antimicrobial activity [28]. MEL-GN and -nGO activity may result from colloid formation (zeta potecial: 23.9 and 36.8 mV), so the complexes were evenly dispersed in the medium with free access to the cells.

The analysis of metabolic activity showed that a significant lowering of this parameter (relative to the MEL itself) was observed in the MEL-nGO treated group for both tested lines. In the case of the MCF-7 line, this parameter was also significantly lower compared to the action of the nanoparticles themselves in the same concentration, which suggests an effective action of the formed complex. ND and MEL-ND complex did not significantly change the metabolic activity of breast cancer cells relative to that in the MEL-treated group. The use of ND at a concentration of 20 mg/L raised this by 5% for MCF-7 cell line. However, using ND with paclitaxel against human lung carcinoma cells resulted in a significant decrease in the metabolic activity of cells compared to the action of the drug itself [29]. Since the drug is an organic compound with a smaller molar mass than MEL, it may be easier for it to interact with the interior of the cell. The decrease in cell viability in the complex-treated groups relative to the MEL treatment alone suggests that MEL may be delivered intracellularly using carbon nanoparticles, causing degradation of membranous structures within the cell leading to their death.

A test checking the integrity of the cell membranes and the outflow of LDH to a medium that, under physiological conditions, is located inside a cell, gave divergent results for cell lines. For MCF-7 only MEL-nGO complex caused a higher level of LDH in the culture medium compared to the MEL-treated group, which implied a stronger destruction of the cell membrane. Interestingly, the nanoparticle itself in the same concentration as in the complex did not show significant differences compared to the control, or to the rest of the studied groups. In turn, for the MDA-MB-231 line, the nGO and MEL-nGO treated group were in the same homologous group as the group treated with 2% triton X solution, which caused 100% membrane degradation. This suggests that MDA-MB-231 is more sensitive to nGO than the MCF-7 cell line. Whereas, the use of ND as MEL carrier for the MDA-MB-231 group resulted in a smaller LDH efflux compared to MEL and a control group. These results indicate that ND can protect MDA-MB-231 cells against the lytic effects of MEL. Study performed with graphitic and oxidized at high pressure and temperature ND on MCF-7 and MDA-MB-231 cell lines demonstrated that MDA-MB-231 cells showed lower uptake of both ND forms that is related to internalization by clathrin-mediated endocytosis and a high expression of clathrin protein in MCF-7 cells [30,31]. Many studies have shown the lytic properties of MEL, also in relation to blood cells. However, the conjugation of MEL with the nanoparticles uptaken by the cells will reduce the spread of MEL across the tissues. Also many studies show that MEL have a non-selective cytolytic effect, but it was proved that MEL selectively degrades cells’ Ras oncogene expression which show overexpressed in 20–50 percent of breast cancers [32,33,34]. Furthermore, studies on normal and lung cancer cells have shown MEL caused lower cytotoxicity to the control human lung fibroblasts cells than to human lung cancer cells [35]. This study suggests that MEL in complexes could have a selective cytotoxic effect on cancer cells, but tests on healthy cells are needed.

For the MDA-MB-231 line, all test groups, except the ND-treated group, caused a significant increase in the level of intracellular ROS. MEL complexes with GN and nGO caused the greatest increase in ROS among all the studied groups for both cell lines. According to scientific reports, ROS has a different role in cancer development depending on the stage of cancer. A decrease in the level of antioxidants and an increase in ROS production supports the process of neoplastic formation in the early stages of cancerogenesis. This induces oxidative damage and mutations in suppressor genes and pro-oncogenes. In advanced stages of cancer, cells often produce high levels of antioxidants to avoid cell death [36]. The results of this test suggest that MEL complexes with nGOs and GN, causing ROS increase, can lead to cell death.

Changes in the potential of mitochondrial membranes are key for the induction of apoptosis. Due to disturbance in the mitochondrial membrane potential of the outer membrane of the transmembrane space protein, especially cytochrome c, a leak occurs into the cytosol, which activates caspase proteases followed by apoptosis [37]. After treatment with MEL-nGO, a significant decrease in mitochondrial membrane potential was observed compared to that in the MEL-treated group. The results suggest that nGO may be an effective MEL carrier to the interior of cells and increases MEL lytic properties toward intrinsic membranes, including the mitochondrial membrane. The increase in cytotoxicity in performed tests in the complex-treated groups suggests that the complexes are taken up by tumor cells and MEL affects intracellular structures. However, for confirmation, the presence of complexes inside the cells should be visualized.

The membrane assay provides information about many proteins while performing the one test, because it allows to study the level of 43 proteins on one membrane. Test results indicate an increase in the level of apoptotic proteins (HTRA, Casp3, Casp8, Bax) and decrease in anti-apoptotic (p21, XIAP) in MDA-MB-231 cells after treatment with MEL-GN. HTRA is a protein secreted by cells in situations of increased stress, for example in fever. It is responsible for the degradation of other proteins, including XIAP, and is considered to be the strongest caspase inhibitor of both apoptosis pathways, mitochondrial and extrinsic [38]. Casp8 is involved in the extrinsic pathway of apoptosis and is responsible for the activation of Casp3. However, Casp3 is an executioner caspase with a critical role in the initiation of apoptosis [39]. Bax protein is responsible for the formation of pores in the outer membrane of the mitochondria, which increases its permeability and accelerates apoptosis process. The p21 protein has many functions, one of which is the inhibition of the apoptosis process [40], therefore lowering its expression may be associated with the start of cell death process. Changes in level of previously described proteins indicates the initiation of apoptosis in MDA-MB-231 cells after MEL-GN treatment. 

Cell death analysis using flow cytometry showed that MEL caused cell death mainly through necrosis. This is consistent with the low level of apoptotic proteins in the MEL-treated group in the membrane array. All examined complexes caused a decrease in necrosis percentage in comparison to the MEL-treated group. The most promising results were obtained for the nGO-MEL complex for both cell lines. The level of necrosis decreased to 12%, while MEL alone caused 26% of necrotic cells, and nGO-MEL caused apoptotic cell growth to 30%, while MEL alone resulted in 20% of apoptotic cells for MCF-7 cell line. Apoptotic cell level was only 9% while necrotic was 37% for the MDA-MB-231 line, but nGO-MEL treated cells had a 22% of apoptotic and 19% for necrotic cells. Apoptosis includes many molecular pathways, generally does not cause inflammation, and is known as a less noxious type of cell death. Interference at any stage of the process can trigger changes in the cells leading to tumor metastasis or resistance to anticancer drugs. On the other hand, necrosis is known as a passive and random cell death without specific mediators. It is an ATP-independent process causing characteristic changes in the cell morphology leading to inflammation, for example, mitochondrial swelling and bursting of cellular membrane. However, studies in recent years suggest that necrosis can also be a relegated phenomenon. Many studies propose the division of necrosis into individual, regulated processes such as necroptosis or mitochondrial permeability transition pore (MPTP)-mediated necrosis [41]. Therefore, the obtained results with a high necrosis do not exclude the potential use of MEL in the treatment of breast cancer. In our previous studies performed on glioblastoma multiforme U87 cell line, we proved that MEL causes cell death mainly through apoptosis and less by necrosis. In turn, studies on ovarian cancer cells showed that MEL induced apoptosis through increased expression of death receptor (DR3, DR4, and DR6) and inhibition of STAT3 pathway. However, studies conducted on MCF-7 and MDA-MB-231 breast cancer lines showed that MEL inhibits the PI3K/Akt/mTOR signaling pathway that may cause autophagy induction [42]. MEL-nGO treated groups had a lower percentage of viable cells and lower level of necrotic cells compared to the control and the MEL-treated group. Activating apoptosis is the most effective way to treat tumors beyond surgery. Currently, the activation of this process is widely studied in relation to many cancers, but in recent years, increasing research has been conducted on programmed necrosis. This is a new trend in research, in particular in cancers that avoid apoptosis. Moreover, performed membrane array show that MDA-MB-231 cells have high levels of Survivin protein, which belongs to the protein family of inhibitor of apoptosis (IAP) and is responsible for caspases inhibition [43]. Therefore, the induction of programmed necrosis may be a solution for the treatment of estrogen-independent breast cancers.

However, this is a preliminary study and further research is needed to find out the exact mechanism of action of melittin and carbon nanoparticle complexes on human breast cancer. If considering the studied complexes as a potential drug in the treatment of breast cancer, we assumed that injection into the tumor would restrict the toxicity of carbon nanoparticles only to the target tissue because carbon nanoparticles taken up by cancer cells limit MEL transport to healthy cells. Nevertheless, the method of administration and the chosen dose might only be relevant for the present model, and administration and effective doses for human treatments must be evaluated in further investigations. Many steps are needed to develop a workable clinical treatment including developing and testing complexes of MEL with carbon nanoparticles dosing strategies to efficiently kill breast cancer and determine which ones have the most innocuous non-specific toxicity and determining ways how these dosing strategies can be used in clinical treatments. To be able to fully relate to the use of subjects in treatment complexes, it is necessary to perform tests on healthy cells and check stability and impact over time of complexes. Also, it is possible that the use of carbon nanoparticles will allow the use of a lower dose of MEL due to the potential internalization of carbon particles by cells. However, the results obtained give grounds for the potential development of the use of these complexes in clinical trials. Research on heathy cells is needed and research that will allow the choice of optimal dose of the complex that will cause toxicity to cancer cells, while non-toxic effects on healthy cells. For the next tests, GN and nGO as MEL carriers were chosen, due to the results consistent with the hypothesis (i.e., they caused an increase in toxicity). The largest differences were observed for MEL-nGO, however, the GN complex reduced the percentage of necrosis relative to MEL alone, therefore it will be included in the study too.

## 5. Conclusions

The results indicate that the MEL complex with nGOs has a stronger toxic effect on breast cancer cells than MEL alone, especially for the estrogen-independent line MDA-MB-231. Moreover, ND protects cells against the lytic effects of MEL. The complexes reduced the level of necrosis compared to MEL but did not completely eliminate it. Carbon nanoparticles can potentially be used as MEL carriers in future treatment. However, further research is necessary to described detailed impact of the complexes at different times and concentration including healthy cells, which will allow determining whether carbon nanoparticles affects the nonspecific effect of MEL.

## Figures and Tables

**Figure 1 materials-13-00090-f001:**
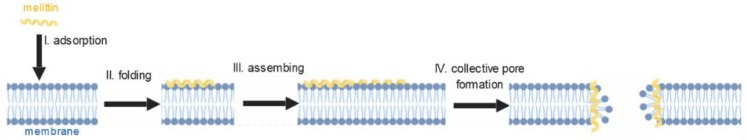
Scheme of pore formation in membrane by melittin.

**Figure 2 materials-13-00090-f002:**
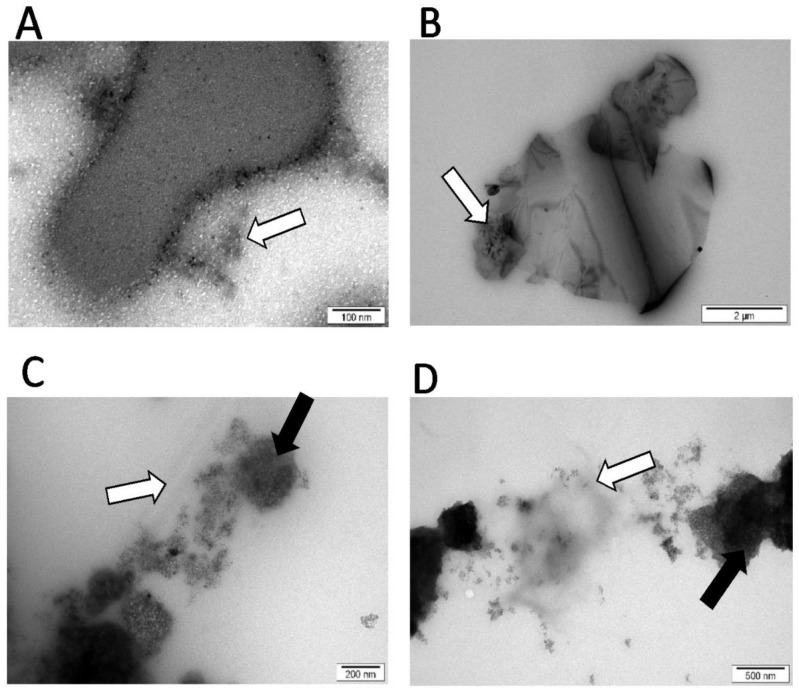
Complexes of melittin and nanoparticles (TEM). (**A**) Melittin (MEL); (**B**) MEL + graphene (GN); (**C**) MEL + nanographene oxide (nGO); (**D**) MEL + nanodiamond (ND). Carbonate nanoparticles = black arrows; MEL = white arrows.

**Figure 3 materials-13-00090-f003:**
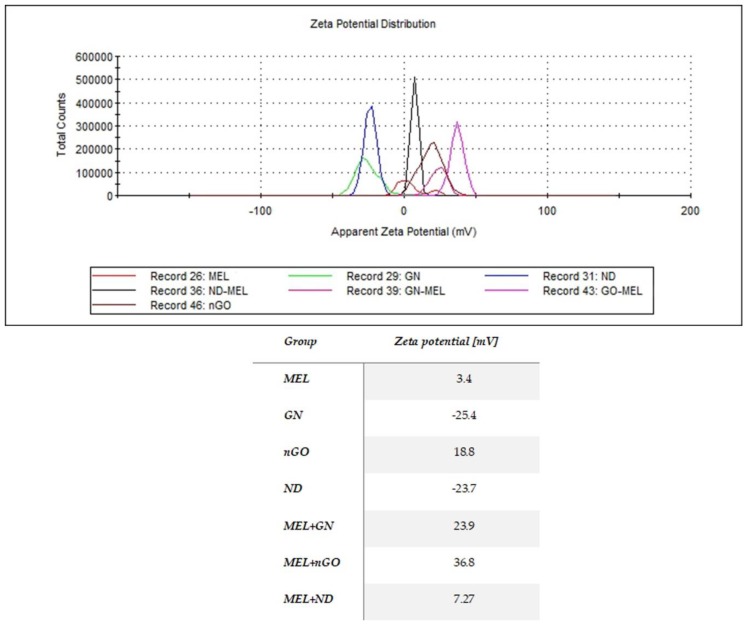
Zeta potential of melittin, nanoparticles and their complexes. MEL—group treated with melittin at a concentration of 10 mg/L; GN—group treated with natural graphene (20 mg/L); Ngo—group treated with graphene nanoxide (20 mg/L); ND—group treated with nanodiamond (20 mg/L); GN + MEL—group treated with melittin complex (10 mg/L) and natural graphene (20 mg/L); nGO + MEL—group treated with melittin complex (10 mg/L) and graphene nanoxide (20 mg/L); ND + MEL—group treated with melittin complex (10 mg/L) and nanodiamond (20 mg/L).

**Figure 4 materials-13-00090-f004:**
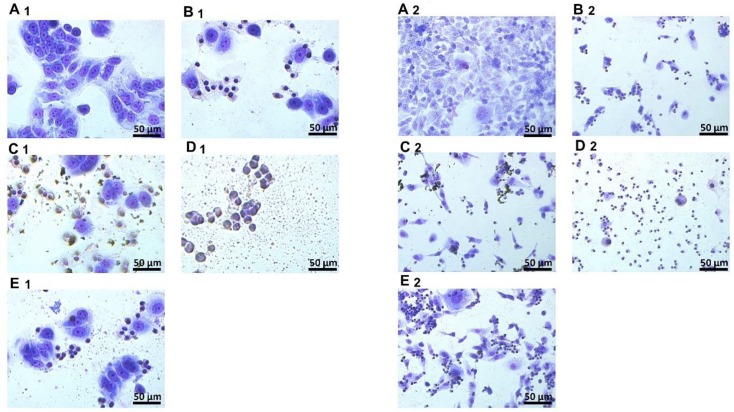
Morphology of MCF-7 and MDA-MB-231 cell lines after MGG staining. A—control group; B—group treated with MEL (10 mg/L); C—group treated with MEL-GN; D—group treated with MEL-GO; E—group treated with MEL-UDD; 1—MCF-7 cell line; 2—MDA-MB-231 cell line.

**Figure 5 materials-13-00090-f005:**
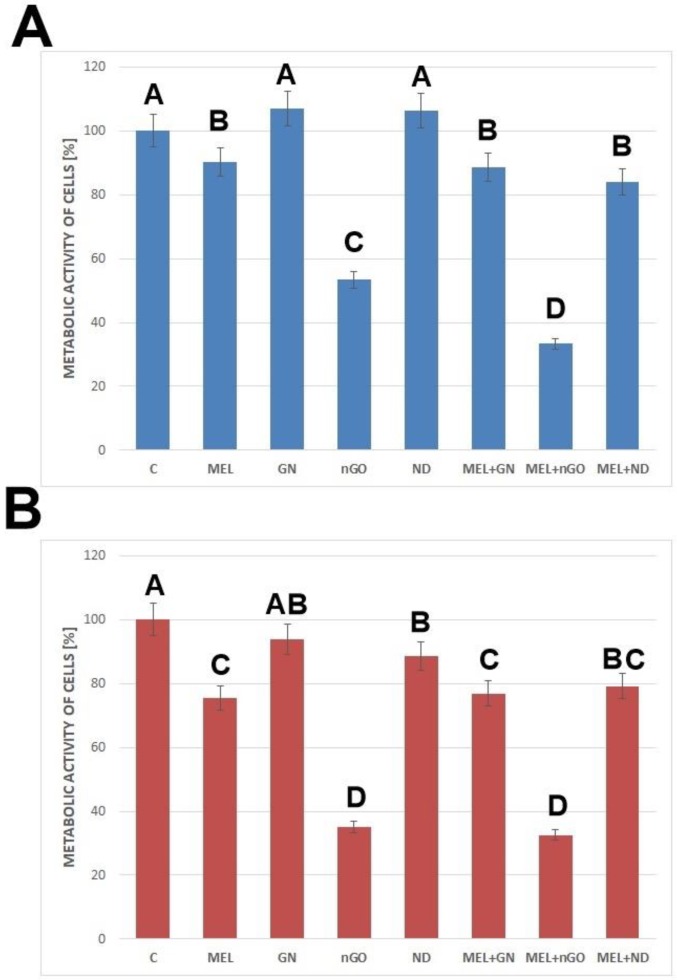
The viability of cells after treatment with melittin complexes with nanoparticles, melittin, and nanoparticles alone. (**A**)— results for MCF-7 cell line, (**B**)—results for MDA-MB-231 cell line; C—control group; MEL—group treated with melittin at a concentration of 10 mg/L; GN—group treated with natural graphene (20 mg/L); nGO—group treated with graphene nanoxide (20 mg/L); ND—group treated with nanodiamond (20 mg/L); GN + MEL—group treated with melittin complex (10 mg/L) and natural graphene (20 mg/L); nGO + MEL—group treated with melittin complex (10 mg/L) and graphene nanoxide (20 mg/L); ND + MEL—group treated with melittin complex (10 mg/L) and nanodiamond (20 mg/L). A, B, C, D—results in one group with different indexes are significantly different (*p*-value < 0.005).

**Figure 6 materials-13-00090-f006:**
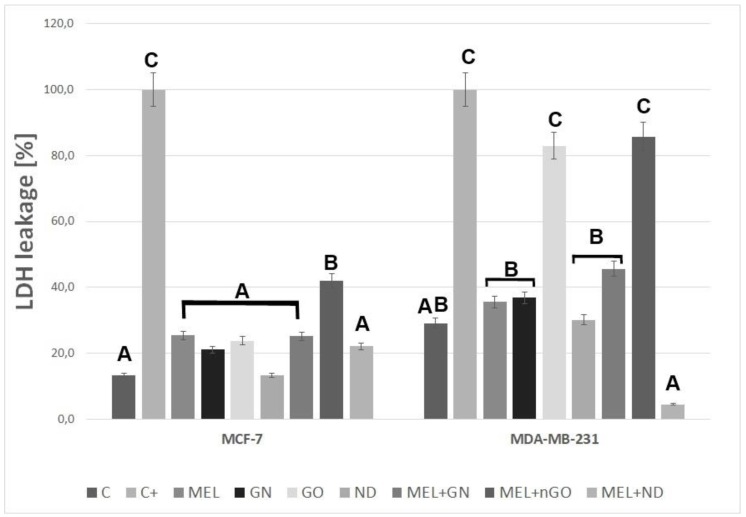
The integrity of MCF-7 and MDA-MB-231 cell membranes after treatment with melittin complexes with nanoparticles, melittin and nanoparticles alone. C—control group; MEL—group treated with melittin at a concentration of 10 mg/L; GN—group treated with natural graphene (20 mg/L); nGO—group treated with graphene nanoxide (20 mg/L); ND—group treated with nanodiamond (20 mg/L); GN + MEL—group treated with melittin complex (10 mg/L) and natural graphene (20 mg/L); nGO + MEL—group treated with melittin complex (10 mg/L) and graphene nanoxide (20 mg / l); ND + MEL—group treated with melittin complex (10 mg/L) and nanodiamond (20 mg/L). A, B, C—results in one group with different indexes are significantly different (*p*-value < 0.005).

**Figure 7 materials-13-00090-f007:**
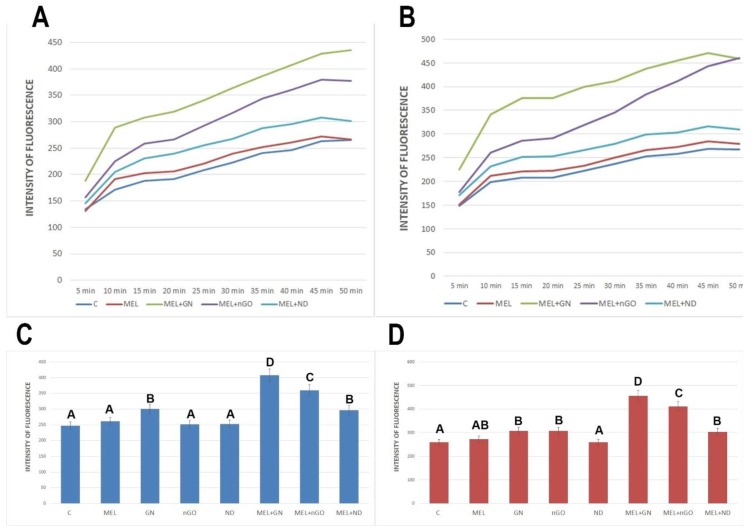
Changes in the level of intracellular reactive oxygen species in MCF-7 and MDA-MB-231 cells after treatment with melittin complexes with nanoparticles, melittin and nanoparticles alone. A,B-Changes in ROS level in time: (**A**)—MCF-7 cell line; (**B**)—MDA-MB-231 cell line. (**C**,**D**)—Changes in ROS level after 40 min after treatment: C—MCF-7 cell line; D-MDA-MB-231 cell line. C—control group; MEL-group treated with melittin at a concentration of 10 mg/L; GN—group treated with natural graphene (20 mg/L); nGO—group treated with graphene nanoxide (20 mg/L); ND—group treated with nanodiamond (20 mg/L); GN + MEL—group treated with melittin complex (10 mg/L) and natural graphene (20 mg/L); nGO + MEL—group treated with melittin complex (10 mg/L) and graphene nanoxide (20 mg/L); ND + MEL—group treated with melittin complex (10 mg/L) and nanodiamond (20 mg/L). A, B, C, D—results in one group with different indexes are significantly different (*p*-value < 0.005)

**Figure 8 materials-13-00090-f008:**
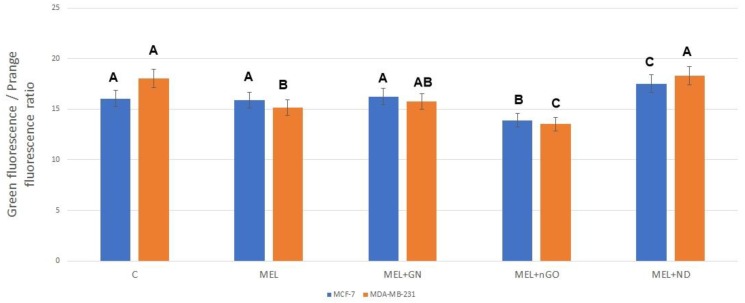
Potential of mitochondrial membranes of MCF-7 and MDA-MB-231 cells after treatment with melittin complexes with nanoparticles and melittin alone. C—control group; MEL-group treated with melittin at a concentration of 10 mg/L; GN + MEL—group treated with melittin complex (10 mg/L) and natural graphene (20 mg/L); nGO + MEL—group treated with melittin complex (10 mg/L) and graphene nanoxide (20 mg/L); ND + MEL—group treated with melittin complex (10 mg/L) and nanodiamond (20 mg/L). A, B, C—results in one group with different indexes are significantly different (*p*-value < 0.005).

**Figure 9 materials-13-00090-f009:**
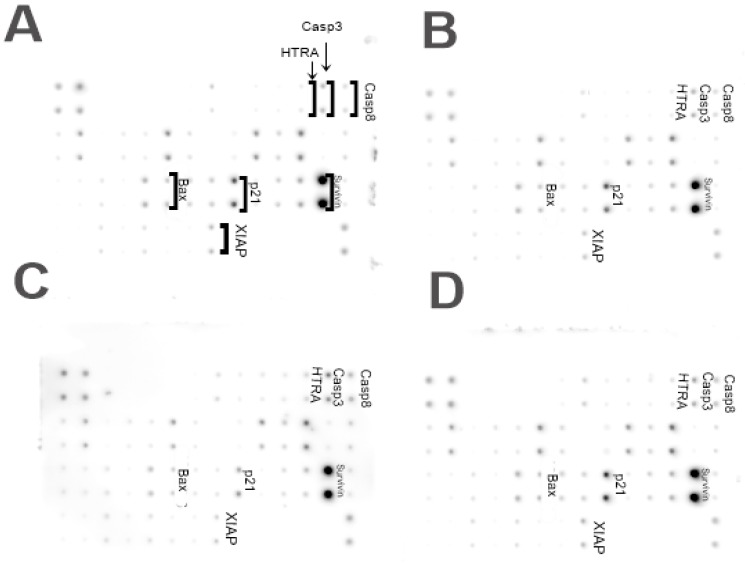
Analysis of the expression of apoptotic proteins of MDA-MB-231 cells. (**A**) Control group; (**B**) group treated with melittin at a concentration of 10 mg/L, (**C**) group treated with a melittin complex (10 mg/L) and natural graphene (20 mg/L); (**D**) group treated with melittin complex (10 mg/L) and nanodiamond (20 mg/L).

**Figure 10 materials-13-00090-f010:**
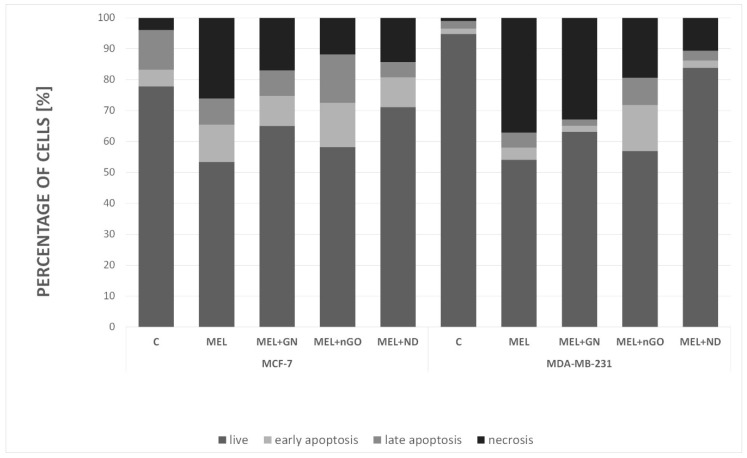
Analysis of cell death type for MCF-7 and MDA-MB-231 cell lines. C—control group; MEL—group treated with melittin at a concentration of 10 mg/L; GN—group treated with natural graphene (20 mg/L); nGO—group treated with graphene nanoxide (20 mg/L); ND—group treated with nanodiamond (20 mg/L); GN + MEL—group treated with melittin complex (10 mg/L) and natural graphene (20 mg/L); nGO + MEL—group treated with melittin complex (10 mg/L) and graphene nanoxide (20 mg/L); ND + MEL—group treated with melittin complex (10 mg/L) and nanodiamond (20 mg/L).

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
