# Peer review of "Use of Selected Carbon Nanoparticles as Melittin Carriers for MCF-7 and MDA-MB-231 Human Breast Cancer Cells"

_materials, 2019, doi:10.3390/ma13010090_

Round 1

Reviewer 1 Report

Although the authors present a really interesting work on the use of carbon nanoparticles for delivery of melittin, there are quite a lot of things that need improvement. More specifically:

Line 102. Authors state that nGO is biocompatible, that is certainly not true and depends on size, concentration, shape among others. While the article cited, is about PEGylated nGO. Line 107. Syntactic errors In the TEM images authors state that melittin is located in between nGO and ND, could you please elaborate more on that? Why is that happening? How is it explained?  Are the Z-pot measurements listed right? nGO has a z-pot of 5.74?? Cell mortality assay are not reported. Authors state that they use Prestoblue assay. A control experiment should be performed without cells, in order to test for reduction of reagents by the nanoparticles, as it is already reported that GO can give false positive results in MTT assay. The presentation of metabolic activity and membrane integrity results are really confusing. It is advisable that separate diagrams for each cell line are constructed and statistically important differences be easier to check. And maybe include another diagram wherever direct comparison of the two cell lines is needed. Why was the analysis of the expression of apoptotic proteins performed only on ND?

Reviewer 2 Report

The manuscript describes the use of carbon nanoparticles, graphene oxide, pristine graphene, and diamond as carriers of melittin in order to increase the toxic effect of melittin to breast cancer. However, the manuscript suffers from lack of novelty, and is clearly to be seen as incremental research. The authors said that the effects of carbon nanoparticles as carriers of melittin on breast cancer cells have not been studied, but all other aspects of the manuscript are lacking novelty, and the results are also not standing out when compared to other similar studies already published. All the figures shown in this manuscript should be improved in order to render the manuscript publishable.

Reviewer 3 Report

The authors describe utilization of graphene oxide, nanodiamond and graphene for delivery of melittin to cancer cells. The nanoparticles were characterized by TEM and light microscope. Cell viability was evaluated as well as membrane integrity.

It is not necessary to write one address nine times. Commercial nanoparticles were used. These were probably well characterized by supplier. It should be fine to present their data to know better what kind of material was used. TEM figures suggest that there is a mixture of mellitin and nanoparticles. How was it proved that melittin is absorbed or binded on nanoparticles? There is no purification method. Zeta potential of nanoparticles and mixtures does not show on high stability of products. It is even worse. Melittin is toxic, what about hemotoxicity test? What is the imagination about targeting breast cancer cells?

Reviewer 4 Report

see file

Round 2

Reviewer 1 Report

Z potential measurements, although they were repeated, still do not seem right. How can nGO have a positive Z-pot? Method of measurement should be provided. The authors state that "The reactions XTT, PrestoBlue, LDH were also carried out without cells  for every tested group and used as a “blank” group". I was referring at the potential of nGO to act as a reducing agent and provide false positive results. Did the authors observe such an effect? And, if so, how was that addressed?

All other comments were properly addressed.

Reviewer 2 Report

Accept 

Author Response

Thanks

Reviewer 3 Report

Suggested changes were made in the text:

GN (thickness: 6-8 nm; purity > 99.5%), nGO (thickness  <2nm; purity 98%; diameter: 8~15um), and ND (thickness 3-4nm; putiry > 95%) powder was purchased from SkySpring Nanomaterials (Houston, TX, USA).

Please, check the text above.

There is no purification method.

Due to the high purity of the materials purchased and the use of ultra-pure water, no purification was carried out:

-melittin – purity ≥ 85%
-GN- purity > 99,5%
-nGO – purity = 98%
-ND – purity > 95%

The question was about removal of unbounded MEL.

Zeta potential of nanoparticles and mixtures does not show on high stability of products. It is even worse.

Thank you for paying attention to this result. The zeta potential measurement was repeated and the following results were obtained.

How is that repeated measurement show different values?

Reviewer 4 Report

see file
